# Decision S4: Efficient Sequence-Based RL via State Space Layers

**Shmuel Bar-David** [*][†]**, Itamar Zimerman** [*][†]**, Eliya Nachmani** [‡][†] **& Lior Wolf** [†]
{shmuelb1,zimerman1}@mail.tau.ac.il
{enk100,liorwolf}@gmail.com

## Abstract

Recently, sequence learning methods have been applied to the problem of off-policy Reinforcement Learning, including the seminal work on Decision Transformers, which employs transformers for this task. Since transformers are parameter-heavy, cannot benefit from history longer than a fixed window size, and are not computed using recurrence, we set out to investigate the suitability of the S4 family of models, which are based on state-space layers and have been shown to outperform transformers, especially in modeling long-range dependencies. In this work we present two main algorithms: (i) an off-policy training procedure that works with trajectories, while still maintaining the training efficiency of the S4 model. (ii) An on-policy training procedure that is trained in a recurrent manner, benefits from long-range dependencies, and is based on a novel stable actor-critic mechanism. Our results indicate that our method outperforms multiple variants of decision transformers, as well as the other baseline methods on most tasks, while reducing the latency, number of parameters, and training time by several orders of magnitude, making our approach more suitable for real-world RL.

## 1 Introduction

Robots are naturally described as being in an observable state, having a multi-dimensional action space and striving to achieve a measurable goal. The complexity of these three elements, and the often non-differentiable links between them, such as the shift between the states given the action and the shift between the states and the reward (with the latter computed based on additional entities), make the use of Reinforcement Learning (RL) natural, see also (Kober et al., 2013; Ibarz et al., 2021).

Off-policy RL has preferable sample complexity and is widely used in robotics research, e.g., (Haarnoja et al., 2018; Gu et al., 2017). However, with the advent of accessible physical simulations for generating data, learning complex tasks without a successful sample model is readily approached by on-policy methods Siekmann et al. (2021) and the same holds for the task of adversarial imitation learning Peng et al. (2021; 2022).

The decision transformer of Chen et al. (2021) is a sequence-based off-policy RL method that considers sequences of tuples of the form (reward, state, action). Using the auto-regressive capability of transformers, it generates the next action given the desired reward and the current state.

The major disadvantages of the decision transformer are the size of the architecture, which is a known limitation in these models, the inference runtime, which stems from the inability to compute the transformer recursively, and the fixed window size, which eliminates long-range dependencies. In this work, we propose a novel, sequence-based RL method that is far more efficient than the decision transformer and more suitable for capturing long-range effects.

The method is based on the S4 sequence model, which was designed by Gu et al. (2021a). While the original S4 method is not amendable for on-policy applications due to the fact that it is designed to train on sequences rather than individual elements, we suggest a new learning method that combines

---

[*]These authors contributed equally to this work.

[†]Tel Aviv University

[‡]Meta AI Research

off-policy training with on-policy fine-tuning. This scheme allows us to run on-policy algorithms, while exploiting the advantages of S4. In the beginning, we trained the model in an off-policy manner on sequences, via the convolutional view. This process exploits the ability of S4 to operate extremely fast on sequences, thanks to the fact that computations can be performed with FFT instead of several recurrent operations. Later, at the fine-tuning stage, we used an on-policy algorithm. While pure on-policy training is a difficult task due to the instability and randomness that arise at the beginning of training, our method starts the on-policy training at a more stable point.

From the technical perspective, our method applies recurrence during the training of S4 model. As far as we can ascertain, such a capability has not been demonstrated for S4, although it was part of the advantages of the earlier HIPPO Gu et al. (2020) model, which has fixed (unlearned) recurrent matrices and different parameterization and is outperformed by S4. Furthermore, in Appendix E we show that the recurrent view of the diagonal state space layer is unstable from both a theoretical and empirical perspective, and we propose a method to mitigate this problem in on-policy RL. This observation provides a further theoretical explanation for why state-space layers empirically outperform RNNs. Moreover, we present a novel transfer learning technique that involves training both the recurrent and convolutional views of S4 and show its applicability for RL.

We conduct experiments on multiple Mujoco (Todorov et al., 2012) benchmarks and show the advantage of our method over existing off-policy methods, including the decision transformer, and over similar on-policy methods.

## 2 RELATED WORK

Classic RL methods, such as dynamic programming(Veinott, 1966; Blackwell, 1962) and Q-learning variants Schwartz (1993); Hasselt (2010); Rummery & Niranjan (1994) are often outperformed by deep RL methods, starting with the seminal deep Q-learning method (Mnih et al., 2015) and followed by thousands of follow-up contributions. Some of the most prominent methods are AlphaGo (Silver et al., 2016), AlphaZero (Silver et al., 2018), and Pluribus (Brown & Sandholm, 2019), which outperform humans in chess, go and shogi, and poker, respectively.

**Sequence Models in RL**     There are many RL methods that employ recurrent neural networks (RNNs), such as vanilla RNNs (Schäfer, 2008; Li et al., 2015) or LSTMs Bakker (2001; 2007). Recurrent models are suitable for RL tasks for two reasons. First, these models are fast in inference, which is necessary for a system that operates and responds to the environment in real-time. Second, since the agent should make decisions recursively based on the decisions made in the past, RL tasks are recursive in nature.

These models often suffer from lack of stability and struggle to capture long-term dependencies. The latter problem stems from two main reasons: (i) propagating gradients over long trajectories is an extensive computation, and (ii) this process encourages gradients to explode or vanish, which impairs the quality of the learning process. In this work, we tackle these two problems via the recent S4 layer Gu et al. (2021a) and a stable implementation of the actor-critic mechanism.

Decision transformers (Chen et al., 2021) (DT) consider RL as a sequence modeling problem. Using transformers as the underlying models, state-of-the-art results are obtained on multiple tasks. DTs have drawn considerable attention, and several improvements have been proposed: Furuta et al. (2021) propose data-efficient algorithms that generalize the DT method and try to maximize the information gained from each trajectory to improve learning efficiency. Meng et al. (2021) explored the zero-shot and few-shot performance of a model that trained in an offline manner on online tasks. Janner et al. (2021) employs beam search as a planning algorithm to produce the most likely sequence of actions. Reid et al. (2022) investigate the performance of pre-trained transformers on RL tasks and propose several techniques for applying transfer learning in this domain. Other contributions design a generalist agent, via a scaled transformer or by applying modern training procedures (Lee et al., 2022; Reed et al., 2022; Wen et al., 2022).

The most relevant DT variant to our work is a recent contribution that applies DT in an on-policy manner, by fine-tuning a pre-learned DT that was trained in an off-policy manner (Zheng et al., 2022).

**Actor-Critic methods**     Learning off-policy algorithms over high-dimensional complex data is a central goal in RL. One of the main challenges of this problem is the instability of convergence, as

shown in (Maei et al., 2009; Awate, 2009; Benhamou, 2019), which is addressed by the Actor-Critic framework (Fujimoto et al., 2018; Peters & Schaal, 2008; Lillicrap et al., 2015; Wang et al., 2016; Silver et al., 2014). It contains two neural components: (i) the critic that parameterizes the value function, and (ii) the actor that learns the policy. At each step, the actor tries to choose the optimal action and the critic estimates its suitability. Similarly to GANs Goodfellow et al. (2014), each neural component maximizes a different objective, and each provides a training signal to the other.

Over the years, several major improvements were proposed: (i) Haarnoja et al. (2018) designed a relatively stable and "soft" version of the mechanism based on entropy maximization, (ii) Mnih et al. (2016) propose an asynchronous variation that achieves impressive results on the Atari benchmark, (iii) Nachum et al. (2017) proposed the Path Consistency Learning (PCL) method, which is a generalization of the vanilla actor-critic system. In this work, we tailored the Actor-Critic-based DPGG algorithm Lillicrap et al. (2015) to implement a stable on-policy training scheme that collects data at the exploration stage and stores it in the replay buffer. This data is then used in an off-policy manner.

**The S4 layer** S4 (Gu et al., 2021a) is a variation of the time-invariant linear state-space layers (LSSL) (Gu et al., 2021b), which was shown to capture long-range dependencies on several benchmarks such as images and videos (Islam & Bertasius, 2022; Gu et al., 2021a), speech (Goel et al., 2022), text Mehta et al. (2022) and more. Recently, an efficient diagonal simplification was presented by Gupta (2022).

## 3 BACKGROUND AND TERMINOLOGY

**The Decision Transformer** formulate offline reinforcement learning as a sequence modeling problem. It employs causal transformers in an autoregressive manner to predict actions given previous actions, states, and desired rewards. Our method adopts this formulation, in which a trajectory is viewed as a sequence of states, actions, and rewards. At each time step $i$, the state, action, and reward are denoted by $s_i, a_i, r_i$, and a trajectory of length $L$ is defined by $\tau := (s_0, a_0, r_0, ..., s_L, a_L, r_L)$. Since the reward $r_i$ at some time step $i$ can be highly dependent on the selected action $a_i$, but not necessarily correlative to future rewards. As a result using this representation for learning conditional actions can be problematic, as planning according to past rewards does not imply high total rewards. Therefore, DT makes a critical design choice to model trajectories based on future desired rewards (returns-to-go) instead of past rewards. The returns-to-go at step $i$ are defined by $R_i := \sum_{t'=i}^{L} r_{t'}$, and trajectories are represented as $(R_0, s_0, a_0, ..., R_L, s_L, a_L)$. At test time, returns-to-go are fed as the desired performance. One central advantage of this formulation is that the model is able to better plan and prioritize long-term and short-term goals based on the desired return at each stage of the episode. For example, the return-to-go is highest at the beginning of the training, and the model can understand that it should plan for long-range. From an empirical point of view, the results of DT showed that this technique is effective for the autoregressive modeling of trajectories.

**The S4 Layer** is described in detail in Appendix A. Below we supply the essential background. Given a input scalar function $u(t)$, the continuous time-invariant state-space model (SSM) is defined by the following first-order differential equation: $\dot{x} = Ax(t) + Bu(t), \quad y(t) = Cx(t) + Du(t)$

The SSM maps the input stream $u(t)$ to $y(t)$. It was shown that initializing $A$ by the HIPPO matrix (Gu et al., 2020) grants the state-space model (SSM) the ability to capture long-range dependencies. Similarly to previous work (Gu et al., 2021a; Gupta, 2022), we interpret $D$ as parameter-based skip-connection. Hence, we will omit $D$ from the SSM, by assuming $D = 0$.

The SSM operates on continuous sequences, and it must be discretized by a step size $\Delta$ to operate on discrete sequences. For example, the original S4 used the bilinear method to obtained a discrete approximation of the continuous SSM. Let the discretization matrices be $\bar{A}, \bar{B}, \bar{C}$.

$$\bar{A} = (I - \Delta/2 \cdot A)^{-1}(I + \Delta/2 \cdot A), \quad \bar{B} = (I - \Delta/2 \cdot A)^{-1}\Delta B, \quad \bar{C} = C \qquad (1)$$

These matrices allow us to rewrite Equation 4, and obtain the recurrent view of the SSM:

$$x_k = \bar{A}x_{k-1} + \bar{B}u_k, \quad y_k = \bar{C}x_k \qquad (2)$$

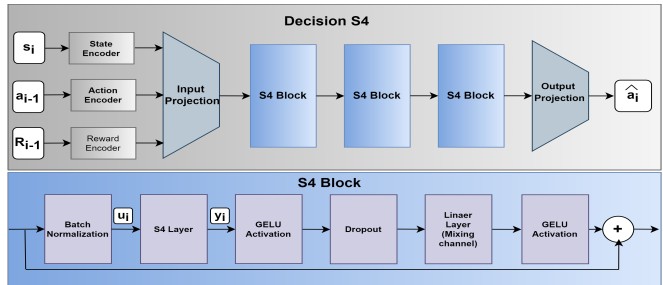

Figure 1: The architecture of the actor network $X$.

Since Eq. 2 is linear, it can be computed in a closed form:

$$y_t = \sum_{i=0}^{t} K_i u_{t-i}, \quad K_i = \bar{C}\bar{A}^{i-1}\bar{B}, \quad y = K * u \tag{3}$$

And this is the convolution view of the SSM, when $*$ denote non-circular convolution.

## 4 METHOD

Below, we describe both an off-policy method, which is motivated by DT and replaces the transformers with S4, and an on-policy method, which applies a tailored actor-critic scheme suitable for S4 layers.

### 4.1 OFF-POLICY

The DT formulation generates actions based on future desired returns. For a trajectory of length $L$, the desired returns ("returns-to-go") are denoted as $R_i = \sum_{i=1}^{L} r_i$, where $r_i$ is the obtained reward on the step $i$. Formally, DT observes a trajectory as $\tau = (s_1, a_1, R_1, \ldots, s_L, a_L, R_L, s_T)$ is a sequence of length $L$ that contains states, actions, and desired total rewards at the moment. Since the transformer has quadratic space and time complexity, running it on complete trajectories is unfeasible, and trajectories were truncated to a length of 20 for most of the environments. While transformer-based variants have achieved excellent results on RL benchmarks, the S4 model is far more suitable for RL tasks for several reasons: (i) The S4 model was shown to be a long-context learner (Goel et al., 2022; Islam & Bertasius, 2022; Gu et al., 2021a), which is relevant to RL (Wang et al., 2020) (ii) RL systems require low latency, and the S4 model is one or two orders of magnitude less computationally demanding during inference than a Transformer with similar capacity. For example, on CIFAR-10 density estimation, the S4 outperforms the transformers and is 65 times faster(see Appendix B Tab. 4) (iii) The number of parameters is much smaller in S4 in comparison to transformers of a similar level of accuracy. These are discussed in great detail in Appendix B.

#### 4.1.1 NETWORK ARCHITECTURE

In the offline setting, a single network $X$ is employed. It consists of three components, as shown in Fig. 1: (i) The S4 component contains three stacked S4 blocks. The internal S4 architecture follows the work of Gu et al. (2021a) and contains a linear layer, batch normalization layer, GELU activations, dropout layer and residual connections. (ii) An encoding layer that encodes the current state, the previous action, and the previous reward. Each encoder is one fully-connected layer with ReLU activations. (iii) The input and output projection layers, each of which contains a single fully-connected layer and ReLu activations. The encoder-decoder components, such as the number of S4 blocks follows the DT architecture.

We used the S4 diagonal parameterization (Gupta, 2022), and set the state size as $N = 64$, similarly to (Gupta, 2022; Gu et al., 2021a), and the number of channels as $H = 64$, half of the DT embedding dimension, since some S4 parameters are complex and consume twice as much memory.

The training scheme of deep reinforcement learning via sequence modeling was extensively explored (Zheng et al., 2022; Kostrikov et al., 2021; Furuta et al., 2021; Chen et al., 2021). Unfor-

tunately, training S4 via this scheme is inefficient, because it involves the S4 recurrent view ( 2) instead of the convolutional view ( 3). Hence, we modify the training scheme to operate on complete trajectories instead of steps.

### 4.1.2 Training Scheme

A sketch of our training scheme is shown in Alg. 1. For simplicity, the sketch ignores batches.

---
**Algorithm 1** Off-Policy Training Scheme

---
1: $X(s_i, R_{i-1}, a_{i-1} \mid \theta_X) = \texttt{Init}()$      ▷ Init the network $X$ according the HIPPO initialization
2: **for** $\tau$ trajectory in dataloader **do**
3:      **for** $i \in [0, ..., L]$ **do**            ▷ Assuming $R_{-1} = R_\tau, a_{-1} = 0$
4:          $\hat{a}_i = X(s_i, R_{i-1}, a_{i-1})$
5:          loss $= |\hat{a}_i - a_i|^2$
6:          Take gradient step on $\theta_X$ according to the loss
7:      **end for**
8: **end for**

---

At each training step a batch of trajectories is randomly drawn from the dataset. The trajectories pad with zeros to reach the same length. Given the current state $s_i$, the previous return-to-go $R_{i-1}$ and the previous action $a_{i-1}$, the network $X$ predicts the next action $\hat{a}_i$. The loss is calculated by applying the $L_2$ loss over a batch of true actions $a_i$ and the predicted ones $\hat{a}_i$.

In comparison to the DT, our scheme operates on complete instead of truncated trajectories. These lead to a much more data-efficient approach that can handle long-context, which, as shown by our experiments, is important for optimal performance. This technique exploits the strengths of S4, and is relevant to any off-policy task. For each task, we select the initial return-to-go $R_{-1}$ from the values $0.5, 0.75, 1.0$, choosing the one that results in the best performance.

### 4.2 On-Policy Fine-Tuning

For on-policy fine-tuning, we present a modified variant of the DDPG actor-critic algorithm (Lillicrap et al., 2015) that includes 4 networks: (i) The actor target $X$, which has the S4-based architecture mentioned above, (ii) the actor copy $X'$ (iii) the critic target $C$, and (iv) the critic copy $C'$. These copy networks are used to achieve soft updates of the actor and critic target networks, using an update factor $\bar{\tau}$ that determines how fast models are updated.

The updated mechanism can be interpreted as using an average of polices. This approach is taken in order to deal with the instability of the actor-critic mechanism, which is a well-known phenomenon, see Sec. 2. Overall, our variant of DDPG has the following properties.

**(i) Training The Critic** To boost stability at the beginning of training, we first freeze the actor and train the critic network alone on the recorded data. Since the purpose of the critic is to estimate the $Q$-function on the policies generated by the actor, we train the critic using the recorded data and not the original dataset. Additionally, we observe that training the actor and critic together from the start increases instability. Both the training of the actor and critic use the replay-buffer to sample batches of transitions. There were instabilities because the actor was changing faster compared to the critic, and also because the critic is not accurate enough before sufficient training iterations when producing gradients for the actor. Those problems that appear when training both models simultaneously required to change the training method. Hence, we trained the actor and critic intermittently, similarly to GANS. In addition, we used a relatively lower learning rate for the actor networks, to prevent situations where the critic falls behind the actor and generates irrelevant reward estimates.

**(ii) Stable Exploration** Another challenge we encountered was maintaining stability while performing efficient exploration. Similarly to other contributions, we limit exploration during training. After the initialization of the critic, the noise is sampled from a normal distribution $N(0, \sigma_{episode})$, where $\sigma_{episode}$ linearly decays over the training from $\sigma$. Additionally, relatively low learning rates are used for all networks; specifically, the critic network used a learning rate of $\alpha_C$, and $\alpha_X$ for the actor.

**(iii) Freezing The S4 Kernel**    On-policy training requires additional regularization, and we regularize the S4 core by freezing $A$. Moreover, in the initial off-policy training, the S4 kernel learns to handle long history via the conventional view, whereas this ability can be lost when training the model via the recurrent view. See Appendixes. B and E for further discussion of the role these two views play.

**Setting Return-to-Go**    At the exploration stage, the trained model produces a replay buffer that contains the explored trajectories. We set the initial return-to-go $R_{-1}$ to a value of $10\%$ higher than the model's current highest total rewards. The initial highest result is estimated by running the trained model with a normalized target of $1.0$. This technique allows us to adjust the initial return-to-go to the recorded data without additional hyper-parameters.

Our on-policy fine-tuning scheme is summarized in Alg. 2. It is based on the Deep Deterministic Policy Gradient (DDPG) algorithm (Lillicrap et al., 2015). As an on-policy algorithm, it contains exploration and exploitation stages. At exploration, the actor collects trajectories and stores them in a replay buffer. We apply the actor-critic framework to these trajectories to narrow the training instability.

---

**Algorithm 2** On-Policy Training Scheme

---

1: Copy the model weights to the actor target network $X'(s_i, R_{i-1}, a_{i-1} \mid \theta_{X'})$
2: Initialize $C(s_i, a_{i-1} \mid \theta_C)$ and a copy $C'(s_i, a_{i-1} \mid \theta_{C'})$
3: Train the critic target network $C(s_i, a_{i-1} \mid \theta_C)$ with respect to $X(s_i, R_{i-1}, a_{i-1} \mid \theta_X)$
4: Apply our on-policy tailored DDPG algorithm in iterations:
5: **for** episode $= 1, M$ **do**
6:     **for** each step in the environment $t$ **do**
7:         Sample noise $w_t \sim N(0, \sigma_{episode})$ where $\sigma_{episode} := \frac{M - episode}{M}\sigma$
8:         Define action $a_t := X(s_i, R_{i-1}, a_{i-1} \mid \theta_X) + w_t$
9:         Play $a_t$ and observe the current reward $r_i$ and the next state $s_{i+1}$
10:         Store $((s_i, a_{i-1}, R_{i-1}), r_i, (s_{i+1}, a_i, R_i))$ in the replay buffer $D$.
11:         Every $K_1$ steps:
12:             Train the critic network $C$ with a relatively high learning rate.
13:             Sample batch from replay-buffer $B = \{(s_i, a_{i-1}, R_{i-1}), r_i, (s_{i+1}, a_i, R_i)\} \subset D$
14:             Calculate Bellman target estimated returns $y = r_i + \gamma C'(s_{i+1}, X'(s_{i+1}, R_i, a_i))$
15:             Update gradient descent for critic: $\theta_C \leftarrow \theta_C - \alpha_C \nabla_{\theta_C}(C(s_i, a_i) - y)^2$
16:             If actor freezing is over, update gradient ascent for actor:
17:                 $\theta_X \leftarrow \theta_X + \alpha_X \nabla_{\theta_X} C(s_i, X(s_i, R_{i-1}, a_{i-1}))$
18:         Every $K_2$ steps:
19:             Soft update the critic target $C'$ from $C$, and the actor target $X'$ from $X$ using $\bar{\tau}$
20:             Update the critic target $\theta'_C \leftarrow (1 - \bar{\tau})\theta'_C + \bar{\tau}\theta_C$
21:             Update the actor target $\theta'_X \leftarrow (1 - \bar{\tau})\theta'_X + \bar{\tau}\theta_X$
22:     **end for**
23: **end for**

---

**Critic Architecture**    The architecture of the actor networks $X, X'$ is the same as the one depicted in Fig. 1. Three fully-connected layers with ReLU activations are used for the critic networks $C, C'$.

## 5    EXPERIMENTS

We evaluate our method on data from the Mujoco physical simulator (Todorov et al., 2012) and AntMaze-v2 (Fu et al., 2020). Additional details about the benchmarks are presented in Appendix C.1. To train the model, we used D4RL (Fu et al., 2020) datasets of recorded episodes of the environments. We trained on these datasets in the form of off-policy, sequence-to-sequence supervised learning.

After training the model, we move on to on-policy training, by running the agent on the environment, while creating and storing in the replay buffer its states, actions, rewards, returns-to-go and also the state of the S4 modules, including those of the next step. A single input for the model can be denoted as $z_i = (s_i, a_i, R_i)$, where $s$ includes both the state provided by the environment, and the states of

Table 1: Normalized Reward obtained for various methods on the Mujoco benchmark. The first segment of the table contains off-line results for non-DT methods, the second segment for DT methods and ours. The bottom part shows results for on-policy fine-tuning. The rows with the $\delta$ present the difference from the corresponding row in the off-line results. Bold denotes the model that is empirically best for that benchmark.

| Dataset: | Hopper | | | HalfCheetah | | | Walker2D | | | Mean |
|---|---|---|---|---|---|---|---|---|---|---|
| Method | Medium | Replay | Expert | Medium | Replay | Expert | Medium | Replay | Expert | Rank |
| IQL (Kostrikov et al., 2021) | 63.81 | 92.13 | 91.5 | 47.37 | 44.1 | 86.7 | 79.89 | 73.67 | **109.6** | 3.66 |
| CQL (Kumar et al., 2020) | 58.0 | 48.6 | **111.0** | 44.0 | 46.2 | 62.4 | 79.2 | 26.7 | 98.7 | 5.22 |
| BEAR (Kumar et al., 2019) | 52.1 | 33.7 | 96.3 | 41.7 | 38.6 | 53.4 | 59.1 | 19.2 | 40.1 | 8.22 |
| BRAC (Wu et al., 2019) | 31.1 | 0.6 | 0.8 | 46.3 | **47.7** | 41.9 | 81.1 | 0.9 | 81.6 | 6.88 |
| Transformer-based methods: | | | | | | | | | | |
| DT (Chen et al., 2021) | 67.6 | 82.7 | 107.6 | 42.6 | 36.6 | 86.8 | 74.0 | 66.6 | 108.1 | 5.55 |
| CDT (Furuta et al., 2021) | 46.6 | – | 80.7 | **52.6** | – | 87.6 | 71.0 | – | 106.7 | 5.66 |
| ODT (Zheng et al., 2022) | 66.95 | 86.64 | – | 42.72 | 39.99 | – | 72.19 | 68.92 | – | 6.16 |
| Sample-based methods: | | | | | | | | | | |
| $TT_u$ (Janner et al., 2021) | 67.4 | **99.4** | 106.0 | 44.0 | 44.1 | 40.8 | 81.3 | 79.4 | 91.0 | 4.33 |
| $TT_q$ (Janner et al., 2021) | 61.1 | 91.5 | 110 | 46.9 | 41.9 | **95.0** | 79.0 | **82.6** | 101.9 | 3.88 |
| **DS4 (ours)** | | | | | | | | | | |
| **DS4** | **89.47** | 87.67 | 110.52 | 47.31 | 43.81 | 94.75 | **81.36** | 80.26 | 109.56 | **2.44** |
| On-policy fine-tuning: | | | | | | | | | | |
| IQL (Kumar et al., 2020) | 66.79 | **96.23** | – | 47.1 | 44.14 | – | 80.33 | 70.55 | – | 2 |
| $\delta$ for IQL | 2.98 | 4.1 | – | 0.04 | 0.04 | – | 0.44 | -3.12 | – | |
| ODT (Zheng et al., 2022) | 97.54 | 88.89 | – | 42.72 | 39.99 | – | 72.19 | 68.92 | – | 2.66 |
| $\delta$ for ODT | 30.59 | 2.25 | – | 0.43 | -0.54 | – | 4.60 | 7.94 | – | |
| **DS4** | **111.3** | 88.42 | – | **47.32** | **45.17** | – | **81.71** | 76.97 | – | **1.33** |
| $\delta$ for **DS4** | 20.9 | 0.03 | – | -0.18 | 0.26 | – | 0.35 | -4.44 | – | |

the S4 kernel. A single replay buffer object can be expressed as $(z_i, r_i, z_{i+1})$. A detailed description of our experimental setup can be found in Appendix C.2

**Results On Mujoco** Tab. 1 lists the results on Mujoco obtained for our method and for baseline methods, both DT-based and other state-of-the-art methods. As can be seen, the our method outperform most other methods on most benchmarks, obtaining the lowest average rank. Compared to DT, which employs sequences in a similar way, we reduce the model size by $84\%$, while obtaining consistently better performance.

Furthermore, our method often outperforms $TT_q$ and $TT_q$, which are sample-based methods that employ beam search to optimize the selected actions over past and future decisions. During inference, these methods run the actor network $PW$ times, where $P$ is the planning horizon and $W$ is the Beam Width, and choose the action that results in the optimal sequence of actions. The results we cite employ $P = 15$, $W = 256$ and the run-time is 17sec in comparison to 2.1ms (see Tab. 4) of our method (4 orders of magnitude).

On-policy results are given for medium and medium-replay. We observe a sizable improvement in hopper-medium, and slight improvement in the other environments. This can be a result of the diminishing returns effect, i.e., the performance of the hopper-medium model was close to the maximal return-to-go already, and the model generated trajectories that are close to this expected reward. In contrast, for the half-cheetah and walker2d, the model finishes the off-policy training phase with rewards significantly lower than out target.

An interesting phenomenon is that for the off-policy performance comparisons, our algorithm appears to be the best when used for the "expert" datasets, rather than "medium" or "medium-replay". We hypothesize that one of the main differences between "expert" and "medium" players is the ability

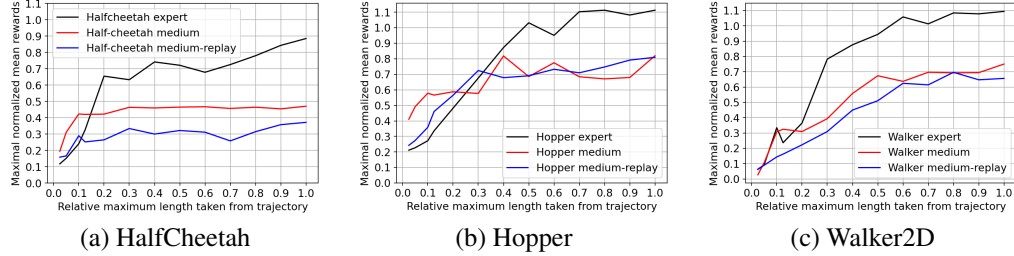

|  |  |  |
| --- | --- | --- |
| (a) HalfCheetah | (b) Hopper | (c) Walker2D |

Figure 2: The effect of limiting the number of recurrent steps while training our model. Maximum mean rewards achieved are presented per environment: (a) HalfCheetah, (b) Hopper (c) Walker2D

to plan for long-range goals. Due to this, the S4 model can fulfill its potential much better in the "expert" case. This hypothesis is supported by Fig. 2, which shows that when the model is enabled to use only a short context ($0.1$ of the maximal context length), all the levels achieved roughly the same performance.

**Results On AntMaze**     Results on AntMaze are shown in Tab. 2. Evidently, in both of the tested environments, DS4 outperforms the other offline method by a large margin. It even outperforms methods that perform online-finetuning (which we did not run yet for DS4 on this benchmark).

Table 2: Normalized Reward obtained for various methods on the AntMaze-v2 benchmark. The baseline results are from (Zheng et al., 2022). Methods that use online-finetunning are denoted by a star ($*$).

| Method: | Transformer-Based | | | IQL | | Ours |
| --- | --- | --- | --- | --- | --- | --- |
| Dataset | DT | ODT | ODT$*$ | IQL | IQL$*$ | **DS4** |
| AntMaze-umaze (Kostrikov et al., 2021) | 53.3 | 53.1 | 88.5 | 87.1 | 89.5 | **90.0** |
| AntMaze-diverse (Kumar et al., 2020) | 52.5 | 50.2 | 56.0 | 64.4 | 56.8 | **79.1** |

**Long-Range Dependencies and The Importance of History**     We also study the importance of S4's ability to capture long-term dependencies. For this, we allow the gradients to propagate only $k$ steps backward. Put differently, the gradients are truncated after $k$ recursive steps.

Fig. 2 presents the normalized reward obtained for various values of $k$, given as a percentage of the maximal trajectory length. Evidently, in all tasks and all off-policy training datasets, as the maximal number of recursive steps drops, performance also drops. This phenomenon is also valid for DT (section 5.3 of (Chen et al., 2021)), supporting the notion that shorter contexts affect the results negatively. Since these benchmarks are fully observable, one might wonder what is the benefit of using any history. The authors of DT hypothesize that the model policy consists of a distribution of policies and the context allows the transformer to determine which policy results in better dynamics.

We would like to add two complementary explanations: (i) The function of recurrent memory is not simply to memorize history. It also allows the model to make future decisions. For example, assume that during evolution a cheetah robot is overturned in a way that has never been seen before on the training data. A smart policy would be to try several different methods, one by one, until the robot overturned again. Namely, recurrent memory can extend the hypothesis class from single to sequential decisions, and the result can improve the robustness of the model. (ii) Even in cases where history does not provide any advantage in terms of expressiveness, it is not clear that the same is true in terms of optimization. Even though all relevant information is included in the state $s_i$, this does not mean that the model can extract all this information. Recurrent models process the previous states in a way that may make this information more explicit.

**Comparisons To RNN**     At test time, we use the recurrent view of S4, and the model functions as a special linear RNN with a transition matrix $\bar{A}$. To measure the impact of the critical S4 components, we compare our model to RNN and practically ablate various components, such as

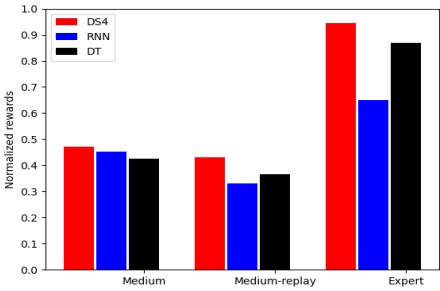

Figure 3: A comparison between S4 and RNN on the HalfCheetah dataset.

HIPPO initialization and the convolutional view (see Sec. B for a discussion of these components). For a fair comparison, only the S4 core model is replaced with RNN, maintaining the DT scheme (for example, using return-to-go), following the technique of training on complete trajectories, and using all non-S4 layers in the internal S4 block (Normalization layers, skip-contentions, and more).

We run these experiments on all the levels of HalfCheetah. The results are depicted in Fig. 3. Evidently, S4 outperforms RNN in all expertise levels, especially in expert datasets. Learning from an expert dataset allows the S4 model to fulfill its potential and learn long-range planning, a skill that cannot be learned at the same level by RNN.

**Low-Budget**    To examine the performance of our model under low-budget scenarios we conducted several additional experiments on all levels of Halfcheetah. While our regular version used $64$ channels ($H$) and an S4 state size of $64$ ($N$), we analyzed how decreasing $N$ and $H$ affect the results.

| Environments | (N=64,H=64) | (N=32,H=64) | (N=16,H=64) | (N=64,H=32) | (N=64,H=16) | (N=32,H=32) | DT |
|---|---|---|---|---|---|---|---|
| Medium | 47.31 | 47.01 | 47.42 | 47.26 | 47.15 | 47.12 | 42.6 |
| Replay | 43.80 | 43.74 | 43.61 | 40.71 | 25.97 | 39.83 | 36.6 |
| Expert | 94.75 | 94.03 | 94.44 | 94.27 | 93.42 | 94.48 | 86.8 |
| % Parameters (Ours) | 1.0 | 0.88 | 0.82 | 0.47 | 0.25 | 0.41 | - |
| % Parameters (DT) | 0.15 | 0.13 | 0.12 | 0.07 | 0.04 | 0.06 | 1.0 |

Table 3: Results of smaller models on HalfCheetah. Each of the smaller models is denoted by (i) $N$ the S4 state size, and (ii) $H$ the number of channels.

Tab. 3 depicts the results for low-budget models. It can be seen that for all levels, even small models ($H = 32$) outperform DT, despite using $93\%$ fewer parameters. In addition, our method doesn't seem sensitive to hyperparameter choices on these tasks. The only drastic decrease in performance occurs when using a tiny model ($N = 64, H = 16$), which uses around $27K$ parameters. Additionally, it can be seen that decreasing the S4 state size $N$ from 64 to 32 or 16 does not affect results, whereas decreasing $H$ from 64 to 32 or 16 negatively affects results in some cases (for example, in the replay environment).

## 6    CONCLUSION

We present evidence that S4 outperforms transformers as the underlying architecture for the (reward,state,action) sequence-based RL framework. S4 has less parameters than the comparable transformer, is low-latency, and can be applied efficiently on trajectories. We also extend the DDPG framework to employ S4 in on-policy RL. While we were unable to train on-policy from scratch, by finetuning offline policies, S4 outperforms the analogous extension of decision transformers.

ACKNOWLEDGMENTS

This project has received funding from the European Research Council (ERC) under the European Unions Horizon 2020 research, innovation program (grant ERC CoG 725974). This work was further supported by a grant from the Tel Aviv University Center for AI and Data Science (TAD). The contribution of IZ is part of a Ph.D. thesis research conducted at Tel Aviv University.

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

# A  BACKGROUND

## A.1  THE S4 LAYER

The S4 layer (Gu et al., 2021a) is a variation of the time-invariant linear state-space layers (LSSL) (Gu et al., 2021b). Below we provide the necessary background on this technology.

**The State-Space Model**    Given a input scalar function $u(t) : \mathbb{R} \to \mathbb{R}$, the continuous time-invariant state-space model (SSM) is defined by the following first-order differential equation:

$$\dot{x} = Ax(t) + Bu(t), \quad y(t) = Cx(t) + Du(t) \tag{4}$$

The model maps the input stream $u(t)$ to $y(u)$. It was shown that initializing $A$ by the HIPPO matrix (Gu et al., 2020) grants the state-space model (SSM) the ability to capture long-range dependencies. Similarly to previous work (Gu et al., 2021a; Gupta, 2022), we interpret $D$ as parameter-based skip-connection. Hence, we will omit $D$ from the SSM, by assuming $D = 0$.

**From Continuous to Discrete SSM**    The SSM operates on continuous sequences, and it must be discretized by a step size $\Delta$ to operate on discrete sequences. For example, the original S4 used the bilinear method to obtained a discrete approximation of the continuous SSM. Let the discretization matrices be $\bar{A}, \bar{B}, \bar{C}$.

$$\bar{A} = (I - \Delta/2 \cdot A)^{-1}(I + \Delta/2 \cdot A), \quad \bar{B} = (I - \Delta/2 \cdot A)^{-1}\Delta B, \quad \bar{C} = C \tag{5}$$

These matrices allow us to rewrite Equation 4:

$$x_k = \bar{A}x_{k-1} + \bar{B}u_k, \quad y_k = \bar{C}x_k \tag{6}$$

This is the recurrent view of the SSM, and it can be interpreted as a linear RNN in which $\bar{A}$ is the state-transition matrix, and $\bar{B}, \bar{C}$ are the input and output matrices.

**Convolutional View**    The recurrent SSM view is not practical for training over long sequences, as it cannot be parallelized. However, the linear time-invariant SSM can be re-written as a convolution, which can be trained much more efficiently. The S4 convolutional view is obtained as follows:

Given a sequence of scalars $u := (u_0, u_1, ...u_{L-1})$ of length $L$, the S4 recurrent view can be unrolled to the following closed form:

$$\forall i \in [L-1] : x_i \in \mathbb{R}^N, \quad x_0 = \bar{B}u_0, \quad x_1 = \bar{A}\bar{B}u_0 + \bar{B}u_1 \quad , \quad ..., \quad x_{L-1} = \sum_{i=0}^{L-1} \bar{A}^{L-1-i}u_i$$

$$y_i \in \mathbb{R}, \quad y_0 = \bar{C}\bar{B}u_0, \quad x_1 = \bar{C}\bar{A}\bar{B}u_0 + \bar{C}\bar{B}u_1 \quad , \quad ..., \quad x_{L-1} = \sum_{i=0}^{L-1} \bar{C}\bar{A}^{L-i-1}\bar{B}u_i \tag{7}$$

Where $N$ is the state size.

Since the recurrent rule is linear, it can be computed in a closed form with matrix multiplication or non-circular convolution:

$$\begin{bmatrix} y_0 \\ y_1 \\ \vdots \\ \vdots \\ \vdots \\ y_{L-1} \end{bmatrix} = \begin{bmatrix} \bar{C}\bar{B} & 0 & 0 & 0 & 0 & 0 \\ \bar{C}\bar{A}\bar{B} & \bar{C}\bar{B} & 0 & \ddots & \ddots & 0 \\ \vdots & \bar{C}\bar{A}\bar{B} & \bar{C}\bar{B} & \ddots & \ddots & 0 \\ \vdots & \ddots & \ddots & \ddots & \ddots & 0 \\ \bar{C}\bar{A}^{L-2}\bar{B} & \ddots & \ddots & \bar{C}\bar{A}\bar{B} & \bar{C}\bar{B} & 0 \\ \bar{C}\bar{A}^{L-1}\bar{B} & \bar{C}\bar{A}^{L-2}\bar{B} & \ldots & \ldots & \bar{C}\bar{A}\bar{B} & \bar{C}\bar{B} \end{bmatrix} \begin{bmatrix} u_0 \\ u_1 \\ \vdots \\ \vdots \\ \vdots \\ u_{L-1} \end{bmatrix}, \tag{8}$$

i.e., $y = \bar{k} * y$ for some kernel $\bar{k}$, which is our convolutional kernel.

**From Theory to Deep Neural Network**   The parameters of each channel of the SSM are $A, B, C, D, \Delta$, and on the forward path $\bar{A}, \bar{B}, \bar{C}$ are computed according to Equation 1. Then, one can use the recurrent view or the convolutional view in the forward path. The SSM is wrapped with standard deep learning components, such as skip-connections, GeLU activations, and normalization layers, and they define the S4 layer(for one channel). While the S4 layer operates on multi-dimensional sequences, the calculations in Equations 6,7 operate on scalars. The S4 layer handles this by stacking multiple copies of the $1-$D layer and using a linear layer to mix information between channels on the same position. To calculate $\bar{k}$ efficiently, structured (Gu et al., 2021a) and diagonal (Gupta, 2022) versions of the state-space model were proposed. We use the latter, which we found to be more stable in our experiments.

## B   MOTIVATION

S4 has a few advantages in comparison to transformers, which make it especially suited for solving RL tasks that are viewed as sequences.

**Long horizon Reward Planner**   Designing RL algorithms for long-term planning is a central goal in RL, which is extensively explored (Ates, 2020; Curtis et al., 2020; Yao et al., 2020; Raposo et al., 2021; Hung et al., 2019). The S4 model was shown, both empirically and theoretically, to excel as a long-context learner. Empirically, S4 achieved SOTA results on every task from the Long Range Arena (Tay et al., 2020), which is a benchmark that consists of several long-context scenarios (images, text, structured data and more). In these experiments, S4 was able to reason over sequences of up to $16,000$ elements. Theoretically, the S4 model includes two well-grounded features that allow it to capture long-range dependencies: (1) **Convolutional View.** The model can be trained via the convolutional view instead of the recursive one. While the recursive view includes $L$ repeated multiplications by $\bar{A}$ (Equation 2), which can result in vanishing gradients, the convolutional view avoids these calculations by using a convolution kernel $\bar{k}$, which can be calculated efficiently and in a stable manner. For example, the original S4 does not compute $\bar{k}$ directly; instead, it applies the inverse FFT on the spectrum of $\bar{k}$, which is calculated via Cauchy kernel and the Woodbury Identity. This is possible due to the Normal Plus Low Rank (NPLR) parameterization of $A$. Furthermore, in contrast to the convolutional view, when using the recurrent view, errors can accumulate during computations, as shown by Oliver (1967). In Appendix E we prove (theoretically) and demonstrated (empirically) that this phenomenon holds for state-space layers, and provide a technique for mitigating those errors during on-policy fine-tuning. (2) **The HIPPO Initialization.** As presented in (Gu et al., 2020) the HIPPO matrix, which defines the S4 initialization and the NPLR-parametrization of the transition matrix $A$, is obtained from the optimal solution of an online function approximation problem. Due to this initialization, the recurrent state is biased toward compressing the representation of the input time series (the history) into a polynomial base. The compressed representation allows the model to handle long history, even with limited state size and relatively few parameters.

To empirically measure the importance of these features for RL tasks, we compare our model to RNN, as described in 3. This ablation study provides empirical support for our main hypothesis, that S4 can be highly relevant for RL tasks, especially in long-range planning. Note that the convolutional view is relevant only for off-policy reinforcement learning, while the HIPPO initialization is relevant to all RL tasks.

**Complexity**   Real-time RL systems require low latency, and the S4 model is one or two orders of magnitude less computationally demanding during inference than a Transformer with similar capacity. For example, on CIFAR-10 density estimation, the S4 outperforms the transformers and is 65x faster. We validate that this phenomenon extends to the RL world, and observe that our method is around 5x faster than DT, as described in Tab. 4. This phenomenon is caused by the fact that the time complexity of the S4 recurrent view does not depend on the sequence length $L$, in contrast to the DT, which is dominated by the $L^2$ factor necessary for self-attention layers. Furthermore, in terms of space complexity, S4 has a critical and promising advantage for real-time and low-budget robots, especially in long-range credit assignments: Note that for these tasks, DT used the entire episode length as the context (section $5.4$ in (Chen et al., 2021)), which resulted in $O(L^2)$ space complexity, making it impractical for long episodes, in contrast to S4, which does not depend on $L$.

| Model | **DS4** | DT($k = 1$) | DT($k = 20$) | DT($k = 100$) | DT($k = 250$) | DT($k = 300$) | $TT_q, TT_u$ |
|---|---|---|---|---|---|---|---|
| Time (ms) | 2.1 | 2.3 | 3.1 | 5.1 | 10.5 | 18.6 | $17K$ |

Table 4: Wall clock running time for 1 step for each model. For DT, we measured the running time for several values of $k$, which is the context length the model takes into account. For TT, we measured two models $TT_q$ and $TT_u$, employing the hyper-parameters that used for Mujoco in Janner et al. (2021). As can be seen, for practical tasks that require long context, S4 can be much more efficient. All experiments were conducted on HalfCheetah environment.

Furthermore, the space and time complexity for the training step of s4 is better than transformers. This gap arises from the fact that the complexity of self-attention layers is quadratic in $L$, while the S4 space complexity depends only on $L$ and the time complexity in $L \log L$. The difference in time complexity arises from the efficiency of the S4 convolutional view, which can be calculated with FFT, compared to the transformer, which requires quadratic complexity. Moreover, at training, the S4 does not materialize the states, causing the time and space complexity to be much more efficient, and, in practice, invariant to the state size.

**Parameter Efficiency** The number of parameters is much smaller in our method in comparison to DT with a similar level of accuracy. For instance, on Hopper we reduce the model size by $84\%$ in comparison to DT. We hypothesize that this phenomenon arises from the recursive structure of S4, which is much more suitable for RL tasks. Since the agent should make decisions recursively, based on the decisions made in the past, RL tasks are recursive in nature, causing non-recursive models such as DT to be inefficient, as they would have to save and handle the history of the current trajectory. Note that this history must be handled even in fully-observable tasks, such as D4RL. For example, the result of DT on the Atari benchmark without history drastically decreased, as shown in Section 5.3 of (Chen et al., 2021).

**Potentially Appropriate Inductive Bias** S4 has a strong inductive bias for recurrent processes (even when trained in a non-recurrent manner) since Equation 2 is recurrent. Also, the HIPPO initialization is designed to implement a memory update mechanism over recursive steps, allowing the model to implement a rich class of recurrent rules that can exploit the compressed history. Regarding RL tasks, we hypothesize that some of the improvement of S4 arises from the suitability of recursive processes for modeling MDPs. MDPs are widely used in the RL literature, both as a theoretical model and as a fundamental assumption employed by various algorithms.

## C    EXPERIMENT DETAILS

### C.1    BENCHMARKS

**Mujoco Benchmark** The bulk of the experiments was done using Deep-mind's Mujoco(Todorov et al., 2012) physical simulation of continuous tasks. The environments we tested our method on were Hopper, HalfCheetah and Walker2D. The offline data per environment was taken from the D4RL(Fu et al., 2020) off-policy benchmark. 3 types of datasets were tested, representing different levels of quality of the recorded data: expert, medium and medium-replay. The expert dataset consists of trajectories generated from a trained actor using the soft actor critic method. The medium dataset is generated from a model trained with the same method, but only halfway through. The medium-replay dataset consists of the replay-buffer data that the medium dataset was training on, on the same time the medium dataset was exported.

**AntMaze Benchmark** The AntMaze navigation tasks benchmark Fu et al. (2020) is designed to measure the model's ability to deal with sparse rewards, multitask data, and long-range planning. In the task, an ant robot learns to navigate a maze. In the umaze task, the ant starts from a fixed point, while in the umaze-diverse it start for a random one. In both tasks, the robot should plan how to reach the target. The reward for this task is binary: as the ant reaches its goal, the reward arises to 1, and otherwise 0. Thus, this task is considered a sparse reward task.

## C.2 EXPERIMENTAL SETUP

For off-policy training we used batches of 32 trajectories, using the maximum trajectory length in that batch as the length for the entire batch, then filling shorter trajectories with blank input. The training was done with a learning rate of $10^{-5}$ and about $10000$ warm-up steps, with linear incrementation until reaching the highest rate. We optimized the model using Adam Kingma & Ba (2014), with a weight decay of $10^{-4}$. The use of the target return of the trajectory per trajectory as input helped make use of less successful trajectories, as seen in previous works, such as the DT variations Chen et al. (2021); Janner et al. (2021). After this training, we continued with on-policy training, using the DDPG method, training the critic first for 35000 steps, and then continuing with both the actor and the critic, but prioritizing the critic in order to preserve stability. This was done by training the actor once for every 3 steps of the critic. We set the target actor and critic update to $\bar{\tau} = 0.1$ to maintain stability.

For on-policy learning, trajectories were generated by targeting the rewards $10\%$ higher than the current model's highest rewards. As mentioned in 4.2, the first estimation is obtained by running the model with a normalized return-to-go of $R_{-1} = 1.0$. As the training continues, we store the highest reward estimate and use it to generate the trajectories. Training for the models was done in batches of 96, with different learning rates for the critic and actor, specifically $\alpha_C = 10^{-3}$ and $\alpha_X = 10^{-5}$. The training occurred every $K_1 = 200$ steps of the environment, and the target models were updated every $K_2 = 300$ steps. To isolate the effects of each on-policy fine-tuning method, we measured the pure contribution of the method to the result. We report these findings in Tab. 1 and denote it by $\delta$. All experiments take less than eight hours on a single NVIDIA RTX 2080Ti GPU, which is similar to DT.

## D   VISUALIZATION OF THE LEARNED DEPENDENCIES

Recall from Eq. 2 and 3 that the S4 recurrent view is $x_k = \bar{A}x_{k-1} + \bar{B}u_k, \quad y_k = \bar{C}x_k$ and it has an equivalent convolution form: $y_t = \sum_{i=0}^{t} K_i u_{t-i}, \quad K_i = \bar{C}\bar{A}^i\bar{B}, \quad y = K * u$.

Recall also that diagonal state-space layers employ $\bar{B} = (1, 1, ..., 1)$ and define $\bar{A}$ as (complex) diagonal matrices which parameterized by $A$ and $\Delta$. We denote the eigenvalues of $\bar{A}$ as $\lambda_j \in \mathbb{C}$ for all coordinate $j \in [N]$. Hence, for a given time $t$ we can write each coordinate $j$ of the recurrent state $x_t(j)$ as $\sum_{i=0}^{t} \lambda_j^i u_{t-i}$. Empirically, we found that $|\lambda_j| \leq 1$ at the end of the training, thus according to De Moivre's formula $|\lambda_j| \geq |\lambda_j^2| \geq ... \geq |\lambda_j^t|$.

The eigenvalues $|\lambda_j|$ control which ranges of dependencies will be captured by $x_t(j)$. For example, in the case that $|\lambda_j| \approx 1$, the history can potentially accumulate over time steps and long-range dependencies can be handled, whereas with $|\lambda_j| << 1$ the information about the history is decaying rapidly over time.

Fig. 4 illustrates how different coordinates handle varying ranges of dependencies and which types of dependencies can be handled by a single coordinate. Coordinates are represented as colors in Fig. 4(a) and as rows in Fig. 4(b). In both cases, it can be seen that different coordinates capture different types of dependencies. For example, since in Fig. 4(a) the red curve decays much faster than the blue curve, it is clear that the red curve targets shorter dependencies.

While Fig. 4 explains how several coordinates of the recurrent state $x_t$ can learn a wide spectrum of dependencies, it ignores the fact that the final output of the SSM depends only on the projection of $x_t$ via $\bar{C}$. Thus, a more accurate view of the learned dependencies must take $K$ into account, which is parameterized by $\bar{C}$.

Recall that in the convolutional view (Eq. 3) the output of the SSM is $y = K * u$ where $K_i = \bar{C}\bar{A}^i$. Therefore, for a given time $t$, the scalar $K_i$ in the kernel controls how the SSM takes the input $u_{t-i}$ into account when calculating $y_t$. Fig. 5 visualizes these kernels and illustrates that they can capture complex and wide dependencies.

Fig. 5(a) depicts how complex dependencies can be expressed by a single kernel. In general, the early part of the blue curve appears to be the most dominant one, which allows the model to handle short dependencies. On the other hand, the last elements of the curve have non-negligible values, thus at least some of the very early history is taken into account. Fig. 5(b) depicts multiple kernels as rows

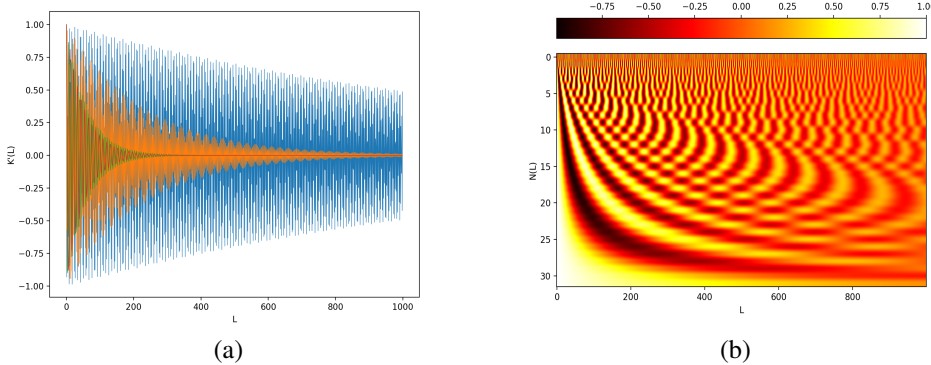

(a) (b)

Figure 4: Various ranges of dependencies are captured by different coordinates of the recurrent state. The x-axis represents the position $i \in [L]$. **(a)** Values of $\lambda_j^i$ are represented on y-axis and colors represent different coordinates.**( b)** Each row of pixels corresponds to specific position $j$ in the recurrent state $x_t(j)$. Each row represents the values of $\lambda_j^i$ along different positions.

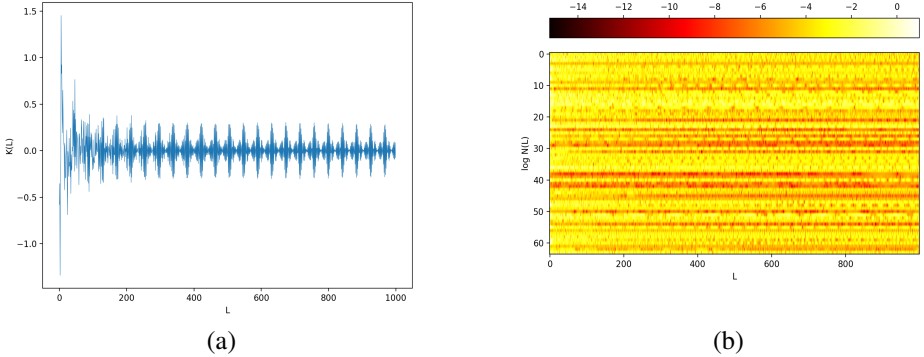

(a) (b)

Figure 5: Various ranges of dependencies are captured by different kernels. The x-axis represents the position $i \in [L]$ in the kernel. **(c)** Values of $K_i$ are represented on the y-axis. **(d)** Each row of pixels corresponds to kernel from separate channels. Each row include a heat map of $K_i$ along $i$, at log-scale.

of a matrix. Evidently, the various kernels capture a wide spectrum of dependencies, and each seems to ignore different parts of the sequence, and target different ranges and frequencies.

# E NUMERICAL INSTABILITY OF THE S4 RECURRENT VIEW

In this section, we prove that the recurrent view of the state-space layer is unstable in some sense. Specifically, the worst-case numerical forward error of a single channel grows proportional in $NL\epsilon_{\textbf{machine}}$, where $L$ is the sequence length and $N$ is the state size (See E.1). We observe that empirically, under the settings in Sec. 5, which contains a sequence of length $L = 1000$, these numerical errors are negligible when the recurrent view is used for inference. In contrast, when the recurrent view is used for training, these errors negatively impact the result. The combination of exposure bias and relatively high numerical errors may explain this phenomenon. Note that when using the convolutional view, the summation can be calculated in a much more stable manner than recursive sum. Specifically, a stable summation can be computed using well-known methods such as Kahan summation algorithm (Kahan, 1965) or 2Sum (Møller, 1965). As far as we ascertain, we are the first to present this theoretical advantage of state-space layers over vanilla RNNs.

## E.1 IDENTIFY THE INSTABILITY

For simplicity, we demonstrate the instability of a real-diagonal state-space layer. Under the assumptions that $\bar{A} = \text{diag}(\lambda_1, ..., \lambda_N)$ and $\bar{B} = (1, ..., 1)$, Eq. 2 can be re-written as:

$$y_t = \bar{C}x_t, \quad x_t(j) = \lambda_j x_{t-1}(j) + u_t \tag{9}$$

Since the obtained computations along the $N$ coordinates of $x_t$ are independent for all $j \in [1, ..., N]$.

Now we can observe the instability of Eq 9. The intuition is that when $\lambda_j x_{t-1}(j)$ and $u_t$ are not on the same scale, this computation can result in loss of significance, which is a well-known phenomenon (Higham, 2002; Goldberg, 1991). Furthermore, by setting $\lambda_j = 1$ in Eq 9, the S4 recurrent view becomes a naive recursive summation algorithm, which is unstable (Muller et al., 2018).

Theorem 1 below shows that the worst-case forward error grows proportional at least as $NL\epsilon_{\textbf{machine}}$, where $L$ is the sequence length and $N$ is the state size.

We start by a standard notation: Evaluation of an expression in floating point is denoted $fl : \mathbb{B} \to \mathbb{B}$, and we assume that any basic arithmetic operations $op$ (for example $'+'$) satisfy:

$$fl(a \quad op \quad b) = (1 + \delta_a^{\text{op}})(a \quad op \quad b), \quad |\delta_a^{\text{op}}| \leq \epsilon_{\textbf{machine}} \tag{10}$$

**Theorem 1.** *Denote the forward error of the recurrent view in Eq. 9 by $\Delta_{s_t}$. $\forall t \geq 0$:*

$$\Delta_{s_t} \leq Nt\epsilon_{\textbf{machine}} \sum_{i=0}^{t} |\lambda_j^{t-i} u_i|$$

**Proof of Theorem 1**

*Proof.* We start by unrolling 9 under the assumption in Eq. 10, and denote the true value of $x_t$ by $sum_t^*$. We denote the actual value of the state under floating-point arithmetic as $sum_t := fl(x_t)$ :

For simplicity, and is not crucial for the proof, we assume that only the '+' operator accumulated errors. Assuming $sum_{-1} = 0$.

$$sum_0 = (1 + \delta_{sum_0}^+)u_0 \tag{11}$$

$$sum_1 = (1 + \delta_{sum_1}^+)\left(u_1 + \lambda_j(1 + \delta_{sum_0}^+)u_0\right) \tag{12}$$

$$sum_2 = (1 + \delta_{sum_2}^+)\left(u_2 + \lambda_j\left[(1 + \delta_{sum_1}^+)\left(u_1 + \lambda_j(1 + \delta_{sum_0}^+)u_0\right)\right]\right) \tag{13}$$

And in general:

$$sum_t = \sum_{i=0}^{t} \lambda_j^{t-i} u_i \Pi_{k=i}^t (1 + \delta_{sum_k}^+) \tag{14}$$

Recall that $|\delta_{sum_k}^+| \leq \epsilon_{\textbf{machine}}$, therefore we can omit factor of $\epsilon_{\textbf{machine}}^2$:

$$\Pi_{k=i}^t (1 + \delta_{sum_k}^+) = 1 + \sum_{k=i}^{t} \delta_{sum_k}^+ + O(t^2 \epsilon_{\textbf{machine}}^2) \tag{15}$$

By plugging it in Eq. 14:

$$sum_t = \sum_{i=0}^{t} \lambda_j^{t-i} u_i \left[ 1 + \sum_{k=i}^{t} \delta_{sum_k}^+ + O(t^2 \epsilon_{\textbf{machine}}^2) \right] \tag{16}$$

And the obtained forward error is:

$$\Delta_{sum_t} := |sum_t - sum_t^*| = \left| \sum_{i=0}^{t} \lambda_j^{t-i} u_i \left( \sum_{k=i}^{t} \delta_{sum_k}^+ + O(t^2 \epsilon_{\textbf{machine}}^2) \right) \right| \leq O(t \epsilon_{\textbf{machine}}) \sum_{i=0}^{t} |\lambda_j^{t-i} u_i| \tag{17}$$

$$\square$$

### E.2 EMPIRICAL EVALUATION OF THE NUMERICAL ERRORS

Fig. 6 examines the forward error across several values of $\lambda_j$, where the input $u_i$ is drawn from a normal standard distribution. The $x$ axis depicts the step number $i$, and the $y$ axis the absolute forward error on this step, averaged over 100 repeats. We used different scales because the rate at which errors accumulate highly depends on $\lambda_j$, as can be expected from theorem 1. We approximate the true solution $sum_i^*$ by computing Eq. 9 under arbitrary precision floating-point arithmetic (2K digits).

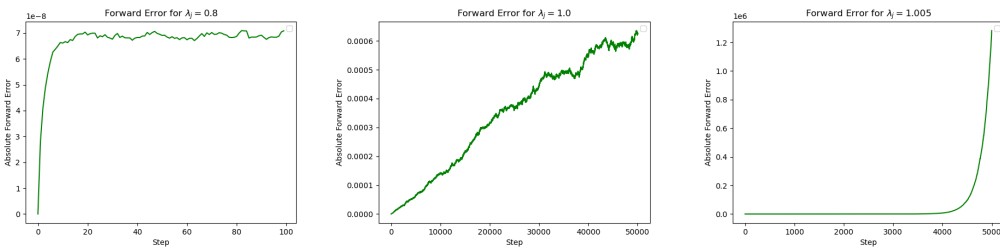

Figure 6: Forward Error for different values $\lambda_j \in \{0.8, 1.0, 1.005\}$

As can be seen from Fig. 6, when $\lambda_j < 1$ the forward error is relatively smaller. However, this is not the important case for two reasons: (i) Where $\lambda_j \approx 1$, long-range dependencies can be handled, since the value of the state $x_t$ is $\sum_{i=0}^{t} \lambda_j^{t-i} u_i$, and when $\lambda_j \approx 1$ the influence of $u_0$ doesn't change over time, allowing the model to copy history between time-steps. (ii) it also the most frequent case. As explain in Appendix D, when $\lambda_j$ is complex, its magnitude $|\lambda_j|$ control the trade-off of which type of dependencies are handled (according to De Moivre's formula). Fig. 7 illustrates that at least at initialization, all the eigenvalues are closed to the unit disk.

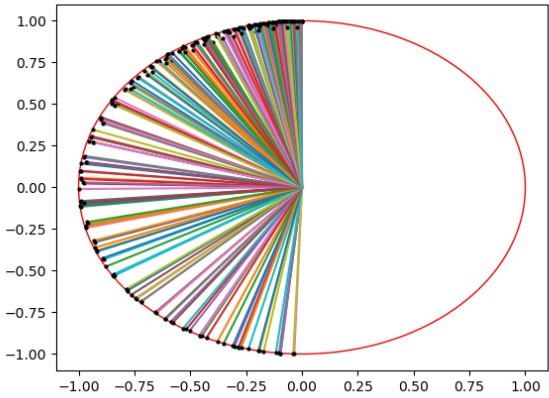

Figure 7: A visualization of the eigenvalues of $\bar{A}$ on initialization, which taken form a diagonal state-space layer that include $H = 10$ channels and state of size $N = 64$. The red circle is the unit disk, and the black points are eigenvalues. Eigenvalues are consistently found in the left part of the plane within and close to the edges of the unit disk.

### E.3  EMPIRICAL EVALUATION ON MUJOCO

We observe that at least on Mujoco benchmark, the forward numerical errors are negligible on inference, and they did not negatively impact the results. When the recurrent view is used for training this is not the case. First, for off-policy learning, we ablate the usage in the convolutional view and train the model via the recurrent view. We found that performance degraded by $0.2\% - 30\%$ in all experiments. To validate that the degradation is not a result of less amount of tuning, we conduct several experiments for each environment over several learning rates, types of optimizer, amount of dropout, and normalization layers. We observe that the gap between using the recurrent or convolutional view is the highest in the medium-replay environments (degradation of $15\% - 60\%$).

Second, for on-policy fine-tuning, we show that freezing the S4 core parameters ($\Delta$ and $A$) mitigates this problem in on-policy RL. The motivation is (i) adding regularization, and (ii) to validate that long-range dependencies captured during the off-policy stage will not diminish. We observe that freezing those parameters (i) increases the stability of the training and decreases the hyper-parameters sensitively. (ii) Accelerate the converges by at least 2x (iii) improve the on-policy results by around $5\%$,

