# OpenReview forum: "Decision S4: Efficient Sequence-Based RL via State Spaces Layers"
_ICLR.cc/2023/Conference — ICLR 2023 poster_

### Official Review · Reviewer_15g8 · 2022-10-24

**Confidence:** 4
**Correctness:** 3
**Technical Novelty And Significance:** 3
**Empirical Novelty And Significance:** 3
**Recommendation:** 6

**Clarity, Quality, Novelty And Reproducibility:**

Quality: solid execution of an conceptually simple idea
Clarity: Structure is good. Writing is ok. Can be improved
Originality: to my knowledge applying S4 to DT-like model is novel.

**Strength And Weaknesses:**

Strengths: Overall I like the motivation of the paper: a lean model that can capture long-range dependency is desirable for any RL agent, and S4 seems to be the perfect backbone for the job. The method proposes a relatively straightforward drop-in replacement of recurrent S4 architecture for the more heavy-duty transformer backbone and shows that the new architecture performs favorably both in task and computational performance.

Weaknesses:
- To me, the offline-online learning scheme is an awkward orthogonal component to the work that’s neither novel nor tightly connected with the core DS4 method. It appears that S4 architecture is not uniquely attuned to online finetuning (compared to transformer / regular RL agents)? And the proposed online finetuning scheme is not tailored to the DS4 architecture either. And the online finetuning is not mentioned in other aspects of evaluation, e.g., long-range dependency and parameter reduction. In the future iteration of this paper, I’d suggest the author remove this component and focus on testing S4 against more challenging domains such as domains that require strong long-term memory.
- The paper claims runtime latency to be a major advantage of S4-based architecture. I'd like to see some empirical results that substantiate this point.
- Minor: the writing needs improvement. Most sentences are way too long for the message they carry.

**Summary Of The Paper:**

The paper presents a method for the recent decision-making-as-sequence-modeling paradigm. It follows the recent trend of treating RL as a sequence modeling problem: given the state-action-reward history, the current state, and the desired reward-to-go, predict the next action. The proposed method is to replace the commonly used transformer model backbone with S4 (Gu et al., 2021) for better long-range sequence modeling capability and reduced model size. The paper also proposes a offline-online learning scheme where the actor is trained with offline data using the decision-transformer-like objective and finetuned online using an actor-critic algorithm.

**Summary Of The Review:**

Overall I think the paper is a solid execution of a conceptually simple idea, although some part of the method is not strictly necessary and may muddle the key message of the paper.

---

> ### Author Response · Authors · 2022-11-08
> **Response to Reviewer 15g8**
>
> We thank the reviewer for the valuable feedback
>
> > The paper claims runtime latency to be a major advantage of S4-based architecture. I'd like to see some empirical results that substantiate this point.
>
> We refer the reviewer to Appendix B, paragraph 3 (complexity) of the original manuscript. This paragraph compares DS4 to DT in terms of time and space complexity, for both inference and training. It is pointed out that:
>
> 1. Empirically, the inference time of DS4 is around 5x faster than DT, as shown in Tab. 4, and several orders of magnitude faster than TT.
>
> 2. Theoretically, the inference time and space complexity of DS4 is independent of the sequence length, while DT has a quadratic dependence. To overcome these limitations, in most of the tasks (except for long-term credit tasks) DT runs on truncated history that has a fixed size of 20, 30, or 50. Therefore, DT can only model relatively short dependencies.
>
> 3. The space and time complexity for one training step of DS4 is better than DT. This gap arises from the fact that the complexity of Transformers layers (which is dominated by self-attention) is quadratic in L, while the S4 space complexity depends only on L and the time complexity in L log L. The difference in time and space complexity arises from the efficiency of the S4 convolutional view, which can be calculated with FFT instead of matrix multiplication.
>
> We believe these three advantages can play a crucial role in real-time RL, especially under low-budget or low-latency constraints.
>
> Furthermore, this empirical gap will increase due to more efficient implementation of the S4 layer and theoretical progress. For example, very recent work shows that training of diagonal state-space layers can be accelerated via Vandemore Multiplication [1], and a new layer called S5 [2], uses parallel scans to reduce the time complexity of the recurrent view.
>
> > the offline-online learning scheme is an awkward orthogonal component to the work that’s neither novel nor … I’d suggest the author remove this component
>
> Online learning plays a crucial role in real-world RL tasks. It is necessary when dealing with small amounts of data or when interacting with a complicated environment. As this work strives to make a paradigm shift, we think it is critical to compare the performance of online fine-tuning DS4 with online fine-tuning DT [3], and show that our method outperforms DT in this domain as well.
>
> >  online finetuning is not mentioned in other aspects of evaluation, e.g., long-range dependency and parameter reduction
>
> We use the same models for online finetuning as used for offline learning, so the parameter efficiency is the same. The obtained model is an improved version of the HiPPO RNN, which still captures long-range dependencies [4].
>
> > It appears that S4 architecture is not uniquely attuned to online finetuning
>
> While most of our online learning techniques are not uniquely attuned for DS4, empirically they work when applied to DS4, and it can be relevant to future works on this domain.
>
> Furthermore, we recommend freezing the S4 kernel in online finetuning, to ensure that long-range dependencies do not diminish because using linear recurrent can accumulate errors during computations [5].
>
> > Writing:
>
> Thank you for this comment. We would send our paper to another round of professional proofreading.
>
> [1] - On the Parameterization and Initialization of Diagonal State Space Models. Albert Gu, Ankit Gupta, Karan Goel, Christopher Ré
>
> [2] - Simplified State Space Layers for Sequence Modeling. Jimmy T.H. Smith, Andrew Warrington, Scott W. Linderman.
>
> [3] - Online Decision Transformer Qinqing Zheng, Amy Zhang, Aditya Grover
>
> [4] - HiPPO: Recurrent Memory with Optimal Polynomial Projections Albert Gu, Tri Dao, Stefano Ermon, Atri Rudra, Christopher Re
>
> [5] - Relative error propagation in the recursive solution of linear recurrence relations. J Oliver

---

> > ### Author Response · Authors · 2022-11-17
> > **Response to Reviewer 15g8 (Cont.)**
> >
> > Dear reviewer 15g8,
> >
> > We hope that the last response in this thread addresses your concerns (runtime latency, offline-online learning). Please also refer to the new content in appendixes C and D. Please let us know if you have any questions or concerns.
> >
> > Thank you,
> >
> > The authors

---

> > > ### Author Response · Authors · 2022-11-18
> > > **Additional Results**
> > >
> > > We have just posted a 3rd revision, which tries to address your suggestion.
> > >
> > > > I’d suggest the author remove this component and focus on testing S4 against more challenging domains such as domains that require strong long-term memory.
> > >
> > > After adding the theoretical results and empirical findings of Appendix D of the 2nd revised manuscript, we consider the on-policy fine-tuning part as an interesting and important contribution. Therefore, we did not remove it.
> > >
> > > Following your suggestion to running on domains that require strong long-term memory, we ran DS4 (off-policy) on the challenging AntMaze benchmark [2] (v2). Due to the time limit, we focus on the environments that were tested by ODT [1] (AntMaze-umaze and AntMaze-umaze-diverse). In the latest revised version, we provide in Tab. 2 a comparison of our results with those of the baseline methods, as reported in [1]. Evidently, our method outperforms the other methods (off-policy and on-policy fine-tuning) by a significant margin. As an example, our method achieves a normalized score that is around 30 points higher than DT and off-policy ODT.
> > >
> > >
> > > [1] - Online Decision Transformer Qinqing Zheng, Amy Zhang, Aditya Grover
> > >
> > > [2] - D4RL: Datasets for Deep Data-Driven Reinforcement Learning. Justin Fu, Aviral Kumar, Ofir Nachum, George Tucker, Sergey Levine

---

### Official Review · Reviewer_1zRF · 2022-10-25

**Confidence:** 4
**Correctness:** 4
**Technical Novelty And Significance:** 2
**Empirical Novelty And Significance:** 4
**Recommendation:** 8

**Clarity, Quality, Novelty And Reproducibility:**

This paper focuses on empirical rather than technical innovation, and from this standpoint is clearly written and positioned. The experiments are extensive and thorough, including many useful ablations analyzing the appropriateness of the proposed method to the problem, as well as the importance of various components of the method.


**Strength And Weaknesses:**

**Strengths:**
- The method is well-motivated. RL can require both efficient batch training as well as efficient step-by-step unrolling, and makes sense as an application of state-space models (SSMs) which can leverage both an efficient parallel training (convolutional mode) and recurrent inference (recurrent mode) of SSMs.
- The on-policy algorithm involves a methodological innovation that involves training with the recurrent view of S4, which was previously only used during inference time.
- The empirical evaluation is comprehensive and the model performs well on standard benchmarks with many established baselines.
- There are extensive empirical ablations, such as on the importance of long-term dependencies, the comparison against more basic RNNs, and on the model size.

**Weaknesses:**
- The methodological innovation is limited. However, the paper is clearly positioned as an empirical work that applies a recent model to a new domain, so this is not a significant weakness.
- It isn't clear how much tuning is required for the proposed method over DT baselines. Also, it isn't clear how much of the proposed algorithms is tailored to DS4; it could be even more compelling if the same algorithms worked for DT but improved with DS4 as a drop-in replacement.



**Summary Of The Paper:**

This paper proposes replacing Learning (RL) methods based on Transformers with the S4 family of models. Two algorithms are proposed for both the on-policy and off-policy setting, showing strong empirical results.


**Summary Of The Review:**

This work applies a recent method (S4) to a new domain (RL), showing promising results in performance and efficiency.

--------
Post-rebuttal: The authors added substantial new content in Appendices explaining several design choices and contrasted the tunability of the model compared to baselines. More improvements have been applied, increasing the performance of the method. All my concerns are addressed and I have raised my score.

---

> ### Author Response · Authors · 2022-11-08
> **Response to reviewer 1zRF**
>
> We thank the reviewer for the valuable feedback.
>
> > It isn't clear how much tuning is required for the proposed method over DT baselines.
>
> It seems that for off-policy learning, the model is not sensitive to hyper-parameters at all. We did not extensively tune the model, and the results could perhaps be improved with additional tuning. For example, we used the state size N and the S4 block architecture (how and which normalization layers were used, types of skip connections) in the same way as the original S4. The number of channels (H), type of optimizer and scheduler, and the overall architecture (encoders, 3 number of blocks) are defined similarly to DT. As shown in Tab. 2 of the submitted manuscript, the performance is relatively stable over different values of H and N.
> For on-policy training, we do notice that the training dynamics is unstable, and there is some sensitivity to the amount of exploration, regularization, and coordination between the actor and critic (which dominates by different learning rates). We consider this instability as a phenomenon that is related to the settings and not to the DS4, since the same S4 model is used.
>
> > It isn't clear how much of the proposed algorithms are tailored to DS4.
>
> For off-policy learning, Alg. 1 follows the basic steps set by DT. There is one major difference: DT operates on truncated sequences (of different lengths), while our algorithm operates on the entire sequence.
> For the on-policy scenario, several design choices were made to deal with the instability of the actor-critic mechanism during training. Based on the problems we encountered, we made several decisions, see Sec. 4.2. Therefore, we assume that algorithm 2 is tailored to DS4 in some ways, but we did not test it with other models instead of DS4 (we do compare it with the DT-based on-policy method ODT).

---

> > ### Comment · Reviewer_1zRF · 2022-11-08
> > **Response**
> >
> > Thanks for the response and the new content! About the new Appendix C, Figure 4(b): I had trouble interpreting this. Is this plotting the actual values of the state $x_t(j)$ where $t$ is indexed by x-axis and $j$ by y-axis, for state size $32$? Or is it just plotting the eigenvalues $\lambda_j^i$?
> >
> > I'm still a little confused about the baselines comparisons. In Table 1, which results are your own reproductions (if any) and which are taken directly from previous papers? If you re-ran any baselines for online RL, were they performed under a similar amount of tuning or attention to details? Section 4.2 is quite involved, which raises the potential concern that DS4 requires more careful consideration for stability that other methods might not need. But I would also believe that this is just inherent to the setting and would apply to all model backbones. Either way, such details are not obvious from the paper. And Table 1 (and the surrounding text) should also make clear which baselines were run by yourselves and which were taken from previous results.
> >
> > Additional (very minor) comments:
> > - Table 1: might make sense to put a horizontal line under Mean to be consistent
> > - Section 3: the 4 in "S4" is not bolded in the paragraph header. Actually it seems that "S4" throughout the paper is produced by a macro which is formatted a little oddly
> > - Appendix B, Table 3: maybe use L instead of k to be consistent with rest of notation?

---

> > > ### Author Response · Authors · 2022-11-16
> > > **Response to reviewer 1zRF (part 1/2)**
> > >
> > > Thanks for the detailed response!
> > >
> > > >  I'm still a little confused about the baselines comparisons. In Table 1, which results are your own reproductions (if any)..?
> > >
> > > All baseline results are taken directly from the previous work (for both on-policy and off-policy training).
> > >
> > > > Section 4.2 is quite involved
> > >
> > > Although our on-policy method is quite involved, it does not seem to be more complex than other similar methods. On-policy IQL[1] used the on-policy training of [15], which employs a unique actor-critic mechanism. ODT[3] uses several novels and advanced techniques to achieve SOTA results such as a combination of standard and custom objectives, applying a relabeling technique on the RTG tokens, and more.
> > >
> > > Furthermore, we believe our scheme is not as involved as it may appear at first. In off-policy training, we use pure imitation learning without estimating or optimizing the Q-function, so when switching to on-policy learning, it is natural to add a critic component that learns the Q-function. This critic architecture is very simple, and consists of three linear layers with ReLU activations, without any normalization layers. Therefore, it makes sense that it requires a different learning rate.
> > >
> > > The 2nd revised manuscript includes a new appendix (D) that explains some of our design choices, such as freezing the S4 kernel during on-policy finetuning.
> > >
> > > > But I would also believe that this is just inherent to the setting and would apply to all model backbones.
> > >
> > > We share this belief with the reviewer. See, Sec.2,4 in ODT [3]:
> > >  “One natural strategy to improve performance is to finetune the pretrained RL agents via online interactions. ***However, the learning formulation for a standard decision transformer is insufficient for online learning***, and as we shall show in our experiment ablations, collapses when used naively for online data acquisition”.
> > > “.. [15] showed that ***naıve application of offline or off-policy RL methods to the offline pre-training and online finetuning regime often does not help, or even hinders, performance*** …”
> > >
> > > > which raises the potential concern that DS4 requires more careful consideration for stability that other methods might not need..
> > >
> > > We do not think that DS4 requires more careful consideration for stability than other methods:
> > > 1. For off-policy training, most of the design choices taken follow previous work (DT[3] or S4[2]), indicating that DS4 is inherently stable.
> > >
> > > 2. We run the on-policy fine-tuning from a sub-optimal start point (5 points lower than off-policy DS4), and observe that the fine-tuning achieves almost the same results (1-2 points less than on-policy DS4).
> > >
> > > 3. The on-policy method employs the off-policy architecture, objective, and training procedure are method. Therefore, we do not adjust\tune the architecture, objective, or training procedure of the off-policy training to improve the on-policy fine-tuning.
> > > 4. Over the years, several tricks were proposed to stabilize the training of Transformers [5-12]. DT[3], and ODT[4] use some of those tricks (such as learning rate warm-up [5], and specific types of normalization layers). In contrast to Transformers, optimizing and designing architectures of state-space layers is still an under-explored territory. With the emergence of more techniques, DS4 would become even more stable. To empirically examine our hypothesis, two very recent improvements are applied to (off-policy) DS4, which we run on all the medium environments: (a) we use different parametrization and initialization for the diagonal state-space layers as proposed in [14]. (b) We initialize the time scale parameter as proposed in [13]. Hooper-Medium improved from 89.47 to 95.2, Cheetah-Medium from 47.32 to 48.9, and Walker-Medium decreased from 81.71 to 81.6. These results indicate that state-space layers have not yet reached their full potential for those benchmarks.
> > >
> > >
> > > >  About the new Appendix C, Figure 4(b):
> > >
> > > The eigenvalues are plotted, not the actual values of the state.
> > >
> > > >   put a horizontal line under …
> > > >  the 4 in "S4" is not bolded in the paragraph …
> > >
> > > The typos have been corrected, thanks for pointing these out!
> > >
> > > > Appendix B, Table 3: maybe use L instead of k..
> > >
> > > The sequence length is denoted as L, like in the original S4 layer paper [2], and the (truncated) context length is denoted as k to correspond as closely as possible to the notations used in DT (for example, see Tab. 5 in [3]).

---

> > > > ### Author Response · Authors · 2022-11-16
> > > > **Response to reviewer 1zRF (part 2/2)**
> > > >
> > > > [1] - Conservative Q-Learning for Offline Reinforcement Learning. Aviral Kumar, Aurick Zhou, George Tucker, Sergey Levine
> > > >
> > > > [2] - Efficiently Modeling Long Sequences with Structured State Spaces. Albert Gu, Karan Goel, Christopher Re
> > > >
> > > > [3] - Decision Transformer: Reinforcement Learning via Sequence Modeling: Lili Chen, Kevin Lu, Aravind Rajeswaran, Kimin Lee, Aditya Grover, Michael Laskin, Pieter Abbeel, Aravind Srinivas, Igor Mordatch.
> > > >
> > > > [4] - Online Decision Transformer. Qinqing Zheng, Amy Zhang, Aditya Grover
> > > >
> > > > [5] - On Layer Normalization in the Transformer Architecture. Ruibin Xiong, Yunchang Yang, Di He, Kai Zheng, Shuxin Zheng, Chen Xing, Huishuai Zhang, Yanyan Lan, Liwei Wang, Tie-Yan Liu
> > > >
> > > > [6] - PowerNorm: Rethinking Batch Normalization in Transformers Sheng Shen, Zhewei Yao, Amir Gholami, Michael W. Mahoney, Kurt Keutzer
> > > >
> > > > [7] - Understanding the Difficulty of Training Transformers. Liyuan Liu, Xiaodong Liu, Jianfeng Gao, Weizhu Chen, Jiawei Han
> > > >
> > > > [8] - Transformers without Tears: ImprovinTransformers without Tears: Improving the Normalization of Self-Attention. Toan Q. Nguyen, Julian Salazarg the Normalization of Self-Attention
> > > >
> > > > [9] - Rethinking Skip Connection with Layer Normalization in Transformers and ResNets Fenglin Liu, Xuancheng Ren, Zhiyuan Zhang, Xu Sun, Yuexian Zou
> > > >
> > > > [10] - PowerNorm: Rethinking Batch Normalization in Transformers Sheng Shen, Zhewei Yao, Amir Gholami, Michael W. Mahoney, Kurt Keutzer
> > > >
> > > > [11] - Improving Deep Transformer with Depth-Scaled Initialization and Merged Attention Biao Zhang, Ivan Titov, Rico Sennrich
> > > >
> > > > [12] - Improving Transformer Optimization Through Better Initialization Xiao Shi Huang, Felipe Perez, Jimmy Ba, Maksims Volkovs
> > > >
> > > > [13] - How to Train Your HiPPO: State Space Models with Generalized Orthogonal Basis Projections. Albert Gu, Isys Johnson, Aman Timalsina,  Atri Rudra, Christopher Re
> > > >
> > > > [14] - On the Parameterization and Initialization of Diagonal State Space Models. Albert Gu, Ankit Gupta, Karan Goel, Christopher Re
> > > >
> > > > [15] - AWAC: Accelerating Online Reinforcement Learning with Offline Datasets. Ashvin Nair, Abhishek Gupta, Murtaza Dalal, Sergey Levine

---

> > > > ### Comment · Reviewer_1zRF · 2022-11-18
> > > > **Thanks**
> > > >
> > > > Thanks for all the clarifications. All my concerns have been addressed and I am raising my score.

---

### Official Review · Reviewer_YrJg · 2022-10-27

**Confidence:** 4
**Clarity, Quality, Novelty And Reproducibility:** Paper is clear, presents a novel appr…
**Correctness:** 3
**Technical Novelty And Significance:** 3
**Empirical Novelty And Significance:** 3
**Recommendation:** 5

**Strength And Weaknesses:**

Strengths:

- To my knowledge this paper presents a novel approach to tackle offline RL
- Applications of the S4/LSSM architectures to control and RL are under-explored and could be beneficial for the community
- S4 does provide almost similar performance to DT with fewer parameters
- Offline-to-online finetuning approach works quite well
- paper is well written

Weaknesses:

- I don't particularly understand why the offline training is capturing long range dependencies, if it is predicting a single action at a time conditioned on the current state and reward to go? Where is this captured? It's important to explain this well.
- How long does it take to train the model vs DT and what is the inference time vs DT?

I would be willing to increase my rating if the questions above are answered.


I would be willing to incre

**Summary Of The Paper:**

This paper presents an offline reinforcement learning approach which is able to capture longer range dependencies than a traditional sequence modeling approaches such as a Decision Transformer (DT). Similar to DT, Decision S4 views RL as a sequence modeling problem, but using the implicit S4 model instead of attention blocks. The offline approach is trained by taking individual transitions and rewards to go, and predicting the actions. The paper also introduces an offline-to-online training approach by first freezing the actor and then the critic, as well as freezing the S4 kernel. Results show that a much smaller model can achieve a similar performance to DT on offline RL tasks.

**Summary Of The Review:**

This paper provides an interesting use of the S4 parameterization and shows that the method works well compared to alternatives (such as Decision Transformer). Details and explanations about what the S4 blocks actually capture in the context of RL are missing which would make the claims in the paper stronger.

---

> ### Author Response · Authors · 2022-11-08
> **Response to reviewer YrJg**
>
> We thank the reviewer for the clarification questions and valuable feedback which helped us to improve the manuscript.
>
> > How long does it take to train the model vs DT and what is the inference time vs DT?
>
> We refer the reviewer to Appendix B, paragraph 3 (complexity) of the original manuscript. This paragraph compares DS4 to DT in terms of time and space complexity, for both inference and training. It is pointed out that:
>
> 1. Empirically, the inference time of DS4 is around 5x faster than DT, as shown in Tab. 4, and several orders of magnitude faster than TT.
>
> 2. Theoretically, the inference time and space complexity of DS4 is independent of the sequence length, while DT has a quadratic dependence. To overcome these limitations, in most of the tasks (except for long-term credit tasks) DT runs on truncated history that has a fixed size of 20, 30, or 50. Therefore, DT can only model relatively short dependencies.
>
> 3. The space and time complexity for one training step of DS4 is better than DT. This gap arises from the fact that the complexity of Transformers layers (which is dominated by self-attention) is quadratic in L, while the S4 space complexity depends only on L and the time complexity in L log L. The difference in time and space complexity arises from the efficiency of the S4 convolutional view, which can be calculated with FFT instead of matrix multiplication.
>
> We believe these three advantages can play a crucial role in real-time RL, especially under low-budget or low-latency constraints.
>
> Furthermore, this empirical gap will increase due to more efficient implementation of the S4 layer and theoretical progress. For example, very recent work shows that training of diagonal state-space layers can be accelerated via Vandemore Multiplication [1], and a new layer called S5 [2], uses parallel scans to reduce the time complexity of the recurrent view.
>
>
> > How long does it take to train the model vs DT?
>
> All experiments take less than 8 hours on a single NVIDIA GPU RTX 2080 Ti, 12K MiB, which is similar to DT, however, DS4 considers a much longer context.
>
>
>
> > understand why the offline training is capturing long range dependencies, if it is predicting a single action at a time conditioned on the current state and reward to go?
>
> The model predicts the next action based on (i) the current environment state, (ii) the reward-to-go, and, most importantly, (iii) the recurrent state of the S4 model.
>
> During inference, the S4 recurrent view, which is expressed in Eq. 2, is used. In offline training, the convolution view is used, and for reasons of efficiency, the recurrent state is not materialized. In this latter case, however, a linear projection of the state is computed via the convolution kernel K (Eq. 3).
>
> In Appendix A in the original manuscript, we provide more information about those two equivalent views. Specifically, long-range dependencies are captured by learning (1) how to compress the history into the recurrent state, and (2) how to predict actions based on the compressed history.
>
>
> > Where is this captured?
>
> The parameters in the layers of DS4 that capture long-range dependencies are A, C, and dt, which control how the recurrent memory is used (Eq. 2), and the types of kernels K (Eq. 3).
> We added visualizations and explanations in the revised manuscript as to how various dependencies are captured, please refer to Appendix C.
>
> [1] - On the Parameterization and Initialization of Diagonal State Space Models. Albert Gu, Ankit Gupta, Karan Goel, Christopher Ré
>
> [2] - Simplified State Space Layers for Sequence Modeling. Jimmy T.H. Smith, Andrew Warrington, Scott W. Linderman.

---

> > ### Author Response · Authors · 2022-11-17
> > **Response to reviewer YrJg (Cont.)**
> >
> > Dear reviewer YrJg,
> >
> > We hope that the last response in this thread addresses your concerns (running time, inference time, where long-range dependencies are captured). Please let us know If there are any remaining concerns.
> >
> > Thank you,
> >
> > The authors

---

### Author Response · Authors · 2022-11-08
**Revised manuscript**

Following the reviews, we are uploading a revised version of the manuscript. Changes are marked in red.

1. We added a new appendix (C) that contains visualizations (Fig. 4 and 5) and explanations of how the S4 parameterization captures multiple types of dependencies.
2. We extended the wall-clock comparison in Tab. 4 (Appendix B). Evidently, S4 has a faster inference time than other DT variants, as the theory predicts.

---

### Author Response · Authors · 2022-11-16
**Notes on the 2nd revision**

Dear reviewers,

Thank you for your extremely helpful feedback, which greatly improved our manuscript.
Following the reviews, we are uploading a revised version of the manuscript. Changes are marked in red.

1. We added a new appendix (D) in the 2nd revision, which contains an empirical and theoretical analysis of the numerical stability of recurrent state-space layers. It explains some of our design choices (freezing the S4 core during on-policy finetuning) and provides an additional novel explanation of the advantages of state-space layers over vanilla RNNs.

2. We fixed several typos. Thank you reviewer 1zRF!

---

### Author Response · Authors · 2022-11-18
**Notes on the 3rd revision**

Following the suggestion of reviewer 15g8, we tested DS4 on a domain with sparser rewards and long-range analysis. We choose the AntMaze benchmark[4] (v2) which is designed to measure the model's ability to deal with sparse rewards, multitask data, and long-range planning.  The comparison of those results with DT[2], ODT[1], and IQL[3] is shown in Tab. 2 of the revised manuscript. As can be seen, our method outperforms the other methods (off-policy and on-policy fine-tuning) by a significant margin.

Additionally, to support reproducibility, we have attached our source code.



[1] - Online Decision Transformer. Qinqing Zheng, Amy Zhang, Aditya Grover

[2] - Decision Transformer: Reinforcement Learning via Sequence Modeling. Lili Chen, Kevin Lu, Aravind Rajeswaran, Kimin Lee, Aditya Grover, Michael Laskin, Pieter Abbeel, Aravind Srinivas, Igor Mordatch.

[3] - Conservative Q-Learning for Offline Reinforcement Learning. Aviral Kumar, Aurick Zhou, George Tucker, Sergey Levine

[4] -D4RL: Datasets for Deep Data-Driven Reinforcement Learning. Justin Fu, Aviral Kumar, Ofir Nachum, George Tucker, Sergey Levine

---

### Decision · Program_Chairs · 2023-01-20

**Decision:**

Accept: poster

**Justification For Why Not Higher Score:**

The novelty can be seen incremental.

**Justification For Why Not Lower Score:**

The result is worth sharing with both the S4 and RL communities.

**Metareview: Summary, Strengths And Weaknesses:**

The paper proposes Decision S4 (DS4) by replacing the Transformer backbone of Decision Transformer by S4. In addition, it also introduces an offline-to-online training method. The experiments are performed on AntMaze-v2 and some MuJoCo tasks (Hopper, HalfCheetah, and Walker2D) and showed that DS4 outperforms DT.

Strength. S4 is a new sequence modeling approach which does not currently have many application success stories yet. So this is good to know that S4 works for offline RL tasks. It is an important knowledge to the community both S4 and RL. Thanks to the efficiency of S4. The proposed method is 5x faster than DT and also requires smaller memory footprint. The paper is clearly well written. Also contains extensive ablations studies. Also in the rebuttal the authors said that it does not require much hyperparameter tuning.

Weakness. The contribution might still seem incremental if we put this work into the framework of simply replacing Transformer in DT by S4. Some reviewers see that the offline-online learning scheme part is an awkward orthogonal component, but I think the author's response makes sense. It's good to see its working in the online setting as well.



**Note From Pc:**

if the above contains the word "oral" or "spotlight" please see: "oral" presentation means -> notable-top-5% and "spotlight" means -> notable-top-25%. As stated in our emails, we are disassociating presentation type from AC recommendations